# Nanostructure Engineering of Metal–Organic Derived Frameworks: Cobalt Phosphide Embedded in Carbon Nanotubes as an Efficient ORR Catalyst

**DOI:** 10.3390/molecules26216672

**Published:** 2021-11-04

**Authors:** Syed Shoaib Ahmad Shah, Tayyaba Najam, Costas Molochas, Muhammad Altaf Nazir, Angeliki Brouzgou, Muhammad Sufyan Javed, Aziz ur Rehman, Panagiotis Tsiakaras

**Affiliations:** 1Hefei National Laboratory for Physical Sciences at the Microscale, CAS Key Laboratory of Soft Matter Chemistry, Department of Chemistry, China University of Science and Technology, Hefei 230026, China; shoaib03ahmad@outlook.com; 2Institute of Chemistry, The Islamia University of Bahawalpur, Bahawalpur 63100, Pakistan; anch39@gmail.com; 3Institute for Advanced Study, Shenzhen University, Shenzhen 518060, China; 4Laboratory of Alternative Energy Conversion Systems, Department of Mechanical Engineering, School of Engineering, University of Thessaly, Pedion Areos, 38834 Athens, Greece; molospao@gmail.com; 5Department of Energy Systems, Faculty of Technology, University of Thessaly, Geopolis, 41500 Larissa, Greece; amprouzgou@uth.gr; 6School of Physical Science and Technology, Lanzhou University, Lanzhou 730000, China; muhammadsj@lzu.edu.cn; 7Laboratory of Materials and Devices for Clean Energy, Department of Technology of Electrochemical Processes, Ural Federal University, 19 Mira Str., 620002 Yekaterinburg, Russia; 8Laboratory of Electrochemical Devices Based on Solid Oxide Proton Electrolytes, Institute of High Temperature Electrochemistry, Russian Academy of Science (RAS), 620990 Yekaterinburg, Russia

**Keywords:** MOFs, nanostructure engineering, heteroatom doping, cobalt phosphide, oxygen reduction reaction

## Abstract

Heteroatom doping is considered an efficient strategy when tuning the electronic and structural modulation of catalysts to achieve improved performance towards renewable energy applications. Herein, we synthesized a series of carbon-based hierarchical nanostructures through the controlled pyrolysis of Co-MOF (metal organic framework) precursors followed by in situ phosphidation. Two kinds of catalysts were prepared: metal nanoparticles embedded in carbon nanotubes, and metal nanoparticles dispersed on the carbon surface. The results proved that the metal nanoparticles embedded in carbon nanotubes exhibit enhanced ORR electrocatalytic performance, owed to the enriched catalytic sites and the mass transfer facilitating channels provided by the hierarchical porous structure of the carbon nanotubes. Furthermore, the phosphidation of the metal nanoparticles embedded in carbon nanotubes (P-Co-CNTs) increases the surface area and porosity, resulting in faster electron transfer, greater conductivity, and lower charge transfer resistance towards ORR pathways. The P-Co-CNT catalyst shows a half-wave potential of 0.887 V, a Tafel slope of 67 mV dec^−1^, and robust stability, which are comparatively better than the precious metal catalyst (Pt/C). Conclusively, this study delivers a novel path for designing multiple crystal phases with improved catalytic performance for energy devices.

## 1. Introduction

Electrochemical energy conversion and storage technologies present great prospects due to their low environmental impact, high energy capacity, and excellent conversion efficiency, etc.; they include metal–air batteries, water electrolyzers, and fuel cells [1,2,3]. For metal-air batteries and fuel cells, the oxygen-reduction reaction (ORR) is considered a significant pathway [4,5,6]. Currently, platinum-based materials are the only available commercial electrocatalysts for efficiently accomplishing the ORR pathway due to their high catalytic performance and stability in alkaline and acidic environments [7,8,9]. However, platinum is an extremely expensive and scarce metal. Therefore, the development of alternative non-Pt ORR electrocatalysts is essential. 

Recently, metal–organic frameworks (MOFs) have attracted extensive interest as novel porous materials [10]. MOFs present a larger surface area, easy functionalization, an enhanced structural diversity and better designability, compared to other traditional porous materials [11]. Additionally, by applying pyrolysis instead of other synthesis routes, MOFs with periodic crystalline structures and porosities can be obtained. This permits the integration of diverse functionalities in a single step as well as the specific control of shape, composition, size, and structure [12,13]. However, most of the synthesis methods used for preparing nanostructures from MOFs require morphology-preserved transformations and a templating strategy [14]. 

Heteroatom-doped MOF precursors can be utilized to synthesize co-doped nanomaterials, e.g., nitrogen-rich zeolitic imidazolate frameworks (ZIFs) [15,16]. This strategy facilitates host–guest interaction and produces more active sites due to geometric and electronic structure modulation. An alternative approach to preparing dual-doped nanomaterials is to introduce the heteroatom precursor into the MOF’s pores [17,18]. For example, Co nanoparticles (NPs) encapsulated in porous carbon structures can be prepared by using Zn-Co MOFs precursors of optimized contents [19]. Furthermore, two MOFs in bimetallic ZIFs (ZIF-67 and ZIF-8) can yield porous carbon polyhedrons due to the high graphitization degree of Co-ZIF (ZIF-67) and the high surface area with more N-atoms of Zn-ZIF (ZIF-8) [20,21,22]. However, pure ZIF-67 derived after pyrolysis usually leads to agglomerated Co NPs with irregular sizes; thus, it could not be considered a potential electrocatalyst for ORR. 

MOF precursors are acknowledged as the best choice for synthesizing catalysts via temperature-programmed pyrolysis with tuned active sites. Additionally, MOF-derived Co_2_P NPs, by using hazardous P-source producing toxic fumes, are established as a bifunctional catalyst due to the formation of various active species, such as N, P co-doped carbon matrix, Co-N_x_, and Co_2_P species [23]. However, the low electronic conductivity and mass transport of pure nanoparticles hinder the ORR performance. The challenge of fabricating nanomaterials with well-controlled structures and regulated physical, chemical, and electronic properties is still remained. 

MOF-derived carbon nanotubes (CNTs) present particular features, such as high surface area, flexibility, and great mechanical strength, facilitating electron and ion transfer [24]. Typically, for the synthesis of 1D MOF-derived nanomaterials, two key approaches are utilized: self-templating and external templating strategies [25,26,27,28]. For example, Te@ZIF-8 was prepared by using an external template (ultrathin Te nanowire), followed by subsequent carbonization to produce hollow carbon nanofibers [29]. In other reports, the CNT growth was achieved by utilizing Fe and Ni metals [30,31]. Accordingly, composite materials doped with Ni, Co, or Fe can be self-templated during pyrolysis to fabricate in situ metal NPs with tubular nanostructures [32]. However, it is challenging to precisely control the dispersion and sites of heteroatoms in CNT-templated MOFs. Therefore, we highlight the urgency to develop facile and eco-friendly strategies for synthesizing MOF-derived catalysts supported on in situ formed Co_2_P, which can synergistically enhance the ORR performance.

Herein, we used an MOF-confined self-template strategy for the structural engineering of Co-coordinated carbon nanotubes and heteroatom-doped Co_2_P sites (P-Co-CNTs) by using phytic acid as an environmentally benign P-source (in contrast to previously reported hazardous P-sources, which cause PH_3_ generation [33,34,35]). Benefiting from the high surface area, the unique facet structure, synergistic effects, and enriched active species, the optimized P-Co-CNT catalyst showed more positive half-wave potential (*E*_1/2_ = 0.887 V vs. RHE), a smaller Tafel slope (67 mV dec^−1^), and robust durability compared to the commercial Pt/C catalyst. The enhanced performance is attributed to the well-dispersed Co-phosphide active sites into the heteroatom-doped CNTs. 

## 2. Results and Discussion

For the first time, we induced electronic structure modulations in Co-CNTs supported on Co_2_P/C matrices. Our MOF-confined synthesis strategy provides highly stable active sites supported on an in situ-produced Co_2_P/C porous support of high surface areas. Figure 1 shows the fabrication of Co-CNTs embedded in the surface of Co_2_P/C porous support. 

The synthesis strategy includes a two-stage carbonization process, which adjusts the electrocatalytic sites and modulates the electronic configuration of the active species during carbonization. During the first carbonization stage, the controlled-temperature pyrolysis leads to the evaporation of the organic linkers leaving Co^2+^ atoms, which steadily aggregate to form Co-CNTs on the NC surface. In the following step, phytic acid is used to remove the inactive species and the P-source, promoting the in situ growth of Co_2_P/C porous/defective support during the second heating. For comparison reasons, catalysts without P-doping (Co-CNTs) and without CNTs growth (Co-NC) were also synthesized and tested.

The scanning electron microscopy images (SEM) confirm the typical morphology of ZIF-67, as shown in Figure 1a. 

After the controlled programmed carbonization of 435/900 °C, CNTs surrounding the polyhedron structures were formed, while the direct carbonization at 900 °C led to the formation of Co NPs over the NC surface (Figure 1b–d). Furthermore, the P-doping of Co-CNTs resulted in the absence of excessive Co NPs, possibly due to the formation of Co_2_P crystal phase in the NC (Figure 1e). Additionally, the porous morphology observed in the SEM image of P-Co-CNT sample indicates the triple role of Co-species as follows: (1) the formation of Co_2_P sites, (2) the prevention of Co agglomeration, and (3) the creation of porosity in the support. 

Figure 1f shows the transmission electron microscopy (TEM) image of Co-CNTs, which clearly reveals the CNT surface embedded with small Co NPs, whereas the sample after P-doping confirms the absence of excessive Co NPs (Figure 1g,h). 

The high-resolution TEM (HR-TEM) image (Figure 1i) further confirms the formation of Co_2_P NPs with a size of 5 nm. Two types of lattice fringes, Co_2_P (121) and CoN_x_ (211) planes, with interlayer spacings of 0.221 nm and 0.22 nm, are observed in P-Co-CNTs. Moreover, as can be seen, the Co_2_P NPs are wrapped inside the graphitic shell, which can efficiently prevent NP corrosion or oxidation during harsh reaction conditions [36]. 

The combination of the scanning transmission electron microscopy and the energy dispersive spectroscopy (STEM-EDS) images (Figure 2a–e) confirms the existence of C, N, P, and Co. Additionally the perfect overlapping of P and Co distributions proves the existence of CoP moieties.

X-ray diffraction (XRD) patterns displayed in Figure 2f demonstrate the typical characteristic peaks of an MOF, confirming the formation of ZIF-67. The diffraction patterns of Co-CNTs and Co-NC samples (Figure 2f and Appendix A) present peaks positioned at 44.3, 51.5, and 76°, consistent with the JCPDS # 15-0806 for Co NPs, indicating their presence in the catalysts. 

Regarding the P-Co-CNT sample, it shows diffraction peaks located at 52.1, 43.3, 40.9, and 40.7°, corresponding to the (002), (211), (201), and (121) planes of Co_2_P (JCPDS # 32-0306). The peaks at 74.98, 50.7, and 43.5° correspond to the (220), (200), and (111) planes of Co-N, respectively (JCPDS # 41-0943). Therefore, the existence of Co_2_P and Co-N planes in P-Co-CNTs is confirmed. 

Raman spectroscopy seen in Figure 3a, carried out to reveal the graphitization degree, shows a higher *I_G_/I_D_* value for the P-Co-CNT than the Co-CNT sample. This result indicates a higher graphitization degree in P-Co-CNTs, which plays a crucial role in boosting the conductivity of the catalyst. The N_2_ adsorption–desorption isotherms displayed in Figure 3b show a significantly higher BET surface area for P-Co-CNTs (1123.6 m^2^/g) than for Co-CNTs (979.1 m^2^/g). This fact indicates that the presence of Co_2_P species in P-Co-CNTs might successively provide adequate active moieties, accelerating the mass transfer during the ORR pathway. 

To further evaluate the surface features of the P-Co-CNT sample, X-ray photoelectron spectroscopy (XPS) was carried out. The XPS survey revealed the existence of C, N, O, P, and Co elements (Figure 3c and Appendix A). The N 1s spectra (Figure 3d) can be fitted into the representative peaks of five kinds of nitrogen, i.e., pyridinic-N, Co-N_x_, pyrrolic-N, graphitic-N, and N-oxide, located at 398.1, 399.4, 399.9, 400.8, and 402.7 eV, respectively. As shown in Figure 3e, the high-resolution Co 2p analysis displays three peaks for Co^0^ (778.4 eV), Co^3+^ (780 eV), and Co^2+^ (782.1 eV), representing the formation of a multifunctional active structure with three kinds of active species. The P 2p spectrum is divided into two main peaks: P-O and P-C (Figure 3f). The existence of the P-O peak demonstrates that the samples were exposed to surface oxidation. Notably, the spin-orbit doublets peaks of P 2p_3/2_ and P 2p_1/2_ observed in the sample are attributed to the presence of the P-Co bond. The O 1s spectrum is fitted in three peaks: CoO_x_, C-O-P/C-O, and C=O (Appendix A). The C 1s spectrum displays the three kinds of peaks: C-O, C-N, and C-C peaks (Appendix A), confirming the successful N-doping of the carbon matrix. Conclusively, all these results confirm the successful fabrication of P-Co-CNTs with homogeneously dispersed Co, Co_2_P, and Co-N_x_ sites embedded in the carbon matrix.

The ORR activity of the as-synthesized catalysts was evaluated in the O_2_-saturated 0.1 M KOH electrolyte by cyclic voltammetry (CV) and linear sweep voltammetry (LSV) measurements on a ring disk electrode (RDE) (Figure 4a and Appendix A). Additionally, Pt/C (20 wt. %) was also tested for comparison. The CV curves reveal a more positive shift in the reduction peak along with greater current density by P-Co-CNTs as compared to Co-CNT catalyst in O_2_-saturated 0.1 M KOH. This is attributed to the reduction in O_2_, since these peaks are absent in CVs conducted in N_2_-saturated solution (Appendix A). Figure 4a shows that the P-Co-CNTs exhibits superior performance, with a more positive half-wave potential (*E*_1/2_ = 0.887 V) than Pt/C (*E*_1/2_ = 0.858 V). 

Notably, an inferior ORR activity was revealed by the un-doped Co-based catalyst (Co-CNTs) with an *E*_1/2_ difference of 71 mV, compared to P-Co-CNTs, thus signifying that after P-doping, the new active species formed improves the intrinsic ORR performance by altering the electronic structure. 

Rotating ring disk electrode (RRDE) tests were conducted to monitor the ORR pathway at 1600 rpm. The electron transfer number for P-Co-CNTs calculated from the RRDE data is around 3.9, indicating a 4e^−^ pathway, thus efficiently catalyzing the oxygen reduction into water (Figure 4b,c). The lowest percentage of peroxide production (1.02%) is exhibited by P-Co-CNTs, whereas Pt/C and Co-CNT catalysts show slightly higher peroxide productions, about 3.7% and 6.2%, respectively (Figure 4b,c). 

The high activity of the P-Co-CNT catalyst is attributed to the high surface areas, the unique facet structures, the superior conductivity, or even to the imprecise synergistic effect of active species. The Tafel slopes were further calculated to evaluate the ORR kinetics (Figure 4d). The superior ORR performance of the P-Co-CNTs becomes apparent from its lower Tafel slope (67 mV dec^−1^) compared to Co-CNTs (98 mV dec^−1^) and Pt/C (69 mV dec^−1^). In order to monitor the role of CNTs towards ORR activity, the performance of a sample fabricated without CNT growth (Co-NC) was investigated. Appendix A shows that the Co-NC catalyst exhibits an *E*_1/2_ negative shift of 67 mV as compared to the Co-CNT catalyst, indicating that the improved ORR activity is due to the vertical growth of CNTs on the NC surface. The electrochemical impedance spectroscopy (EIS) (Figure 4e) demonstrates the lowermost polarization (charge-transfer and mass-transfer) resistance for P-Co-CNTs compared with the un-doped catalyst, due to the well-defined channels (CNTs) and the P-doping that can form new active species, improving the intrinsic ORR performance by altering the electronic structure. 

Since the stability of the catalysts is crucial for real-world utilization, we investigated the stability of the as-synthesized catalysts at a sweep rate of 50 mV s^−1^ for 8000 CV-cycles in O_2_-saturated 0.1 M KOH solution. Figure 4f shows that the P-Co-CNTs electrocatalyst exhibits strong long-term stability, while almost retaining its initial activity, by showing a minimum loss in *E*_1/2_ of 2 mV. On the contrary, the Pt/C catalyst shows a higher activity decay of 10 mV in *E*_1/2_. In conclusion, we confirmed that P-Co-CNT is a promising ORR electrocatalyst with respect to other reported catalysts (Appendix A). 

Based on the above findings, we summarized that the structural changes occurring through P-doping promote the formation of Co_2_P species. Phosphorous has a higher electron donation ability, larger atomic radius, and similar chemical properties with respect to nitrogen. Nitrogen can induce a net positive charge on carbon atoms due to the difference between their electronegativity values [37]. Correspondingly, P-doping may facilitate the ORR process by positively inducing charge to the carbon atoms, promoting the attraction of electrons and the effective breaking of the O−O bonding [38,39,40,41]. In addition, we confirmed the crucial role of CNTs on the improved ORR activity by demonstrating a lower ORR activity for a catalyst synthesized without CNTs growth (Appendix A) [21]. In conclusion, the superior ORR catalytic activity of P-Co-CNTs is attributed to the synergistic effect between the single Co-CNTs, Co_2_P species, and the P- and N-doped carbon matrix. 

## 3. Materials and Methods

### 3.1. Materials 

Cobalt nitrate hexahydrate, 2-methyl imidazole, phytic acid, and methanol were purchased from Aladdin supplier with 99.9% purity. 

### 3.2. Catalyst Preparation 

#### 3.2.1. Synthesis of ZIF-67 

Solution A was prepared by homogeneously dissolving Co(NO_3_)_2_·6H_2_O (3 g) in methanol (60 mL). Then, Solution B, containing 45 mmol of 2-methylimidazole (2-MeIM) and dispersed in 60 mL of methanol, was mixed with solution A and continuously stirred for 24 h. The final product was collected by centrifugation and washed several times with methanol. 

#### 3.2.2. Synthesis of Co-CN

A sample of 0.5 g of ZIF-67 was placed in a ceramic boat and carbonized at 900 °C under Ar/H_2_ (95:5) environment at 5 °C/min for 3 h.

#### 3.2.3. Synthesis of Co-CNTs

The as-obtained ZIF-67 sample was carbonized, firstly at 435 °C for 1 h under an Ar/H_2_ (95:5) environment with a rate of 3 °C min^−^^1^, and secondly, the temperature further increased to 900 °C for 3 h with 5 °C min^−^^1^.

#### 3.2.4. Synthesis of P-Co-CNTs

Phytic acid (4 mL) was added into 20 mL of ethanol containing homogeneously dispersed Co-CNTs (100 mg), followed by 12 h of stirring. Then, the final product was collected after washing with deionized water and dried in an oven. The final sample was obtained after carbonization under an Ar/H_2_ (95:5) environment with 5 °C min^−^^1^ at 900 °C for 1 h.

### 3.3. Physicochemical Characterization

The morphology of the as-synthesized samples was monitored by SEM, TEM, HR-TEM and STEM-EDS. The scanning electron microscopy (SEM) images were recorded on JSM-7800F (JEOL) at an accelerated voltage of 15 kV. The transmission electron microscopy (TEM), high-resolution TEM (HR-TEM), and EDS (energy dispersive spectroscopy) images were observed on Tecnai G2F20S-TWIN at an accelerated voltage of 200 kV. Furthermore, the crystal structure of the samples was monitored by X-ray diffraction (XRD, Shimadzu X-ray diffractometer) at a scan rate of 5 °C min^−^^1^. The N_2_ adsorption–desorption isotherms were collected on Kubo X1000 sorption analyzer instrument at 77 K, and specific surface areas were estimated by the Brunauer–Emmett–Teller (BET) method. Raman spectroscopy was carried out on LabRamHR evolution spectrometer operated at 532 nm, equipped with a Nb-Yag laser excitation source. The elemental composition and valence states were studied by X-ray photoelectron spectroscopy (XPS, ESCALAB250Xi spectrometer). 

### 3.4. Electrochemical Characterization 

The Autolab electrochemical workstation was used to carry out all the ORR tests. The electrochemical measurements were conducted in a three-electrode system, consisting of a working electrode (a glassy carbon rotating disk), a counter electrode (a graphitic carbon rod), and a reference electrode (vs. Ag/AgCl). The recorded working potentials vs. Ag|AgCl (3M KCl) were converted into the reversible hydrogen electrode (RHE) according to the equation E(RHE) = E (Ag|AgCl) + 0.059 pH + 0.210 V, where pH (0.1 M KCl) = 5.8 [42]. A catalyst loading of 0.3 mg cm^−2^ was deposited on the glassy carbon electrode surface (0.19625 cm^2^). Cyclic voltammetry (CV) and linear sweep voltammetry (LSV) curves were taken in O_2_-saturated (or N_2_-saturated) 0.1 M KOH solution at sweep rates of 50 mV s^−1^ and 10 mV s^−1^, respectively. 

The rotating ring disk electrode (RRDE) experiments for the ORR pathway study were performed in O_2_-saturated solution in a standard three-electrode cell at room temperature. The total electron-transfer number (n) and the hydrogen peroxide yield (%H_2_O_2_) were determined from RRDE tests, as seen below: (1)$$n=\frac{4\times I_{D}}{I_{D}+\frac{I_{R}}{N}}\;$$
(2)$$\%\;Hydrogen\;peroxide=\left(\frac{I_{D}+\frac{I_{R}}{N}}{I_{D}+\frac{I_{R}}{N}}\right)\times 100$$
where *N* (=0.37) is the H_2_O_2_ collection efficiency of the Pt ring, *I_D_* is the disk current, and *I_R_* is the ring current.

Electrochemical impedance spectroscopy was carried out at 0.8 V vs. RHE for ORR within the frequency range of 10^5^ and 0.01 Hz and an AC potential amplitude of 10 mV. 

## 4. Conclusions

In summary, the successful fabrication of the Co-CNTs and Co_2_P embedded in an N-doped carbon matrix was achieved via a structural modulation strategy. The heteroatom-doped nanostructured porous carbon materials with high surface areas, permanent and uniform porosity, and well-regulated functionalities were attained, which would be difficult using conventional methods. The optimized P-Co-CNT catalyst showed a higher *E_1/2_* of 0.887 V vs. RHE compared to Co-CNTs and Pt/C (20 wt.%) catalysts. These findings showed that the heteroatom doping of Co-CNTs in the carbon framework generates charge redistribution and synergistic effects, resulting in improved catalytic performance. Hence, this strategy paves a new route for improving heterogeneous catalysis toward energy applications.

## Data Availability

The data presented in this study are available on request from the corresponding authors.

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
