# Peer review of "Nanostructure Engineering of Metal–Organic Derived Frameworks: Cobalt Phosphide Embedded in Carbon Nanotubes as an Efficient ORR Catalyst"

_molecules, 2021, doi:10.3390/molecules26216672_

Round 1

Reviewer 1 Report

This article provides an excellent example of thoughtful and productive research in the field of nanochemistry. The method proposed by the authors for the preparation of a new type of catalyst will undoubtedly find wide application in the future. The text presented in the article does not cause serious remarks. The article in question can be recommended for publication in the Molecules as presented.

Author Response

Answer: Thank you very much for your professional evaluation. The authors are grateful for accepting our article without any changes.

Reviewer 2 Report

The authors report on metal nanoparticles embedded in carbon nanotubes which show superior activity toward ORR compared to commercial Pt/C (20%). Notably, the reported catalyst suppress the formation of H2O2 as side product which is known as source for membrane degradation in PEM fuel cells. Hence, this work merit publication in this journal. Few minor comments need to be addressed prior to publication:

  • all LSV curves and performance data need to be reported with the angular velocities of the RRDE in the Figure captions, i.e. at 2000 rpms.
  • The authors use Ag|AgCl as reference electrode during the experiment, but the working potentials in the LSV curves and CVs are vs. RHE. Please add the conversion relationship "E(RHE) = E(Ag|AgCl) + 0.059*pH + E0(Ag|AgCl) V and cite ref. https://doi.org/10.3390/polym10091002. E0(Ag|Cl) is 0.210 V for Ag|AgCl filled with 3 M KCl and 0.235 V when filled with 1 M KCl. pH of 0.1 M KOH is 13.5.

Author Response

Question 1: All LSV curves and performance data need to be reported with the angular velocities of the RRDE in the Figure captions, i.e. at 2000 rpms.

Answer 1: Thank you very much for your kind suggestion. We have added the rpms in the figure captions.

Question 2: The authors use Ag|AgCl as reference electrode during the experiment, but the working potentials in the LSV curves and CVs are vs. RHE. Please add the conversion relationship "E(RHE) = E(Ag|AgCl) + 0.059*pH + E0(Ag|AgCl) V and cite ref. https://doi.org/10.3390/polym10091002. E0(Ag|Cl) is 0.210 V for Ag|AgCl filled with 3 M KCl and 0.235 V when filled with 1 M KCl. pH of 0.1 M KOH is 13.5.

Answer 2: We are thankful for your constructive comment. We have added the suggested change in the manuscript and also cited the suggested reference.